# Synthesis of Thermo-Responsive Block-Graft Copolymer Based on PCL and PEG Analogs, and Preparation of Hydrogel via Click Chemistry

**DOI:** 10.3390/polym11050765

**Published:** 2019-05-01

**Authors:** Pei Shang, Jie Wu, Xiaoyu Shi, Zhidan Wang, Fei Song, Shouxin Liu

**Affiliations:** Key Laboratory of Applied Surface and Colloid Chemistry, Ministry of Education, School of Chemistry and Chemical Engineering, Shaanxi Normal University, Xi’an 710119, China; sp13772442791@163.com (P.S.); 18375847342@163.com (J.W.); 18647847370@163.com (X.S.); 18189608643@163.com (Z.W.); 17563713257@163.com (F.S.)

**Keywords:** *ε*-Caprolactone, block-graft copolymer, click chemistry hydrogel

## Abstract

Thermo-responsive cross-linkable *m*PEG-*b*-[PCL-*g*-(MEO_2_MA-*co*-OEGMA)]-*b*-*m*PEG was synthesized by ring-opening polymerization (ROP) and atom transfer radical polymerization (ATRP). Then, the cross-linkable block-graft copolymer was used to prepare hydrogel via a copper-catalyzed 1,3-dipolar azide-alkyne cycloaddition reaction. The chemical structure and composition of copolymer were characterized by proton nuclear magnetic resonance (^1^H NMR), Fourier-transform infrared (FT-IR) and gel permeation chromatography (GPC). The self-assembly behaviors of the copolymer in aqueous solution were studied by UV spectrophotometer, fluorescence probes, the surface tension method, dynamic light scattering, and transmission electron microscopy. The results proved that the copolymer has excellent solubility and better temperature response. The three-dimensional network structure of the gels, observed by scanning electron microscopy at different temperatures, indicated that the gels have temperature response.

## 1. Introduction

An environmentally responsive amphiphilic copolymer, composed of hydrophilic and hydrophobic groups, cannot only self-assemble into core–shell micelles in aqueous solution, but produce correspond alter according to small changes in surrounding, so this copolymer has great potential application in drug delivery [1,2]. Hydrogels with three-dimensional network structure have attracted increasing attention because of their unique structure and exceptional properties [3,4,5]. However, the poor mechanical strength and non-biodegradability of common hydrogels dampen their application in biomedical fields. As is well known, the content of rigid monomer in the comonomer directly affects the properties of amphiphilic polymer; it also influences the swelling and mechanical properties of hydrogel [6]. Therefore, it is necessary to synthesis an environmentally responsive amphiphilic copolymer with suitable monomers, and then to prepare hydrogel by the amphiphilic copolymer [7,8]. The special swelling and mechanical properties of the responsive hydrogel, decide that the environmentally responsive hydrogel could be a novel material for many subareas of biomedical, such as drug carriers for controlled-release, tissue engineering, gene delivery, artificial muscles, etc. [9,10,11,12,13,14,15].

There are numerous ways to prepare hydrogels, such as free radical polymerization, microemulsion polymerization, “click chemistry” [14], and the like. While, the radical polymerization method has a low monomer conversion rate, and hydrogels produced in this way showed a disordered molecular structure [15]. Microemulsion polymerization has a limited application range [16]. The Cu(I)-catalyzed dipolar cycloaddition reaction of azide-alkyne groups, named “click chemistry”, is one of the most common methods for synthesizing gel because it can produce various polymers at low-cost [17,18,19]. In particular, the catalyst of click chemistry is nontoxic to cells, the reaction can be carried out in water or organic solvents, and the speed of the gel formed can be easily controlled. All of the above determined that click chemistry is an ideal method for preparing a three-dimensional network hydrogel [20,21].

It is well known that polyethyleneglycol (PEG) and poly (*ε*-caprolactone) (PCL) are widely used in the biomedicine field because of the superior properties that nontoxic, nonimmune, and biodegradable [22,23,24,25,26,27,28]. PCL with higher initial strength is used as a toughening biomaterial in bone repair or tissue engineering due to its slow degradation rate and mechanical properties. But, a disadvantage of PCL is its water solubility, which limits the application of PCL–PEG copolymers [29,30]. The PEG analog copolymer was formed by the polymerization of 2-(2-methoxy ethoxy)ethyl methacrylate (MEO_2_MA) and oligoethylene glycol methyl methacrylate (OEGMA), except it had excellent water solubility similar to PEG, also owning to its unique temperature sensitivity. Moreover, the molar ratio between MEO_2_MA and OEGMA added to the reaction can affect the low critical solution temperature (LCST) of the copolymers [31,32,33]. Built on the above conditions, this work designed to synthesize a cross-linkable *m*PEG-*b*-[PCL-*g*-(MEO_2_MA-*co*-OEGMA)]-*b*-*m*PEG block-graft copolymer, which contains MEO_2_MA and OEGMA monomers to improve the water solubility and temperature sensitivity properties.

In addition, two different cross-linkers were synthesized in this work. The well-defined cross-linker P(GMA-*co*-MEO_2_MA-*co*-OEGMA) contained 2-(2-methoxy ethoxy) ethyl methacrylate (MEO_2_MA) and oligoethylene glycol methyl methacrylate (OEGMA), with the same good water solubility and temperature sensitivity as the *m*PEG-*b*-[PCL-*g*-(MEO_2_MA-*co*-OEGMA)]-*b*-*m*PEG block-graft copolymer. TPOM is another cross-linker, which has multiple cross-linking points [34]. Then, two kinds of hydrogels with different internal three-dimensional network densities were formed via click chemistry. The two hydrogels have great potential value in the biomedical field.

## 2. Materials and Methods

### 2.1. Materials

*ε*-Caprolactone (99%), monomethoxy poly(ethylene glycol) (*M*_W_ = 750 g·mol^−1^), and cuprous chloride were purchased from Alfa Aesar (Shanghai, China). Stannous octoate (Sn(Oct)_2_, 96%), Glycidyl methacrylate (GMA, 99%), Pentaerythritol (98%), and Tetrabutylammonium bromide were purchased from J&k (Beijing, China). Hexamethylene diisocyanate(HMDI), 2-(2-methoxy ethoxy) ethyl methacrylate (MEO_2_MA), and oligoethylene glycol methyl methacrylate (OEGMA) were purchased from TCL. Propargyl bromide, *N*-dodecyl mercaptan, and *N*, *N*, *N*′, *N*′, *N*″-pentamethyl diethylenetriamine (PMDETA) were purchased from Macleans (Shanghai, China). Sodium azide was purchased from Zhengzhou Penny Chemical Reagent Factory (Zhengzhou, China), 2,2′-bipyridine was purchased from Guangdong Guanghua Technology Co., Ltd. (Guangdong, China). Dichloromethane, *n*-hexane, anhydrous diethyl ether, toluene, methanol, *N*, *N*-dimethylformamide (DMF), 1,4-dioxane, acetone, and other reagents were purchased from Sinopharm Chemical Reagent Co., Ltd. (Shanghai, China). Among them, *ε*-Caprolactone should be distilled under reduced pressure situation after being dried over CaH_2_ for 48 h at room temperature. All solvents were distilled before used. Methylene chloride was refluxed over CaH_2_for 4 h, and then distilled. Toluene was refluxed over sodium for 4 h and then distilled for use.

### 2.2. Synthesis

#### 2.2.1. Synthesis of Block Copolymer mPEG-b-PCL-b-mPEG

The synthesis of the triblock copolymer *m*PEG-*b*-PCL-*b*-*m*PEG used the method of ring-opening polymerization (ROP). The specific operations were appropriate amount of *m*PEG, and a certain molar ratio of *α*-Chloro-*ε*-caprolactone (*α*Cl*ε*CL) and *ε*-Caprolactone (*ε*CL) were taken in a completely dry three-necked flask and dissolved in toluene under a stirring bar. Three drops of catalyst stannous octoate were added to the system, the reaction was heated at 120 °C in an oil bath under the protection of nitrogen atmosphere. After moderate stirring for 12 h, the reaction system was cooled to 60 °C. Hexamethylene diisocyanate (HMDI) was added into the mixture and reacted with the diblock copolymer *m*PEG-*b*-PCL in a ratio of 1:2 for another 6 h. Then, the reaction system was cooled to room temperature. Isolated the product by precipitated in cold methanol, then dissolved in dichloromethane, and isolated by precipitated in cold diethyl ether again. The *α*-chloro-*ε*-caprolactone used in this step was synthesized according to the method of the literature [35]. ^1^H NMR (CDCl_3_): *δ*-1.68 (m, 3H), *δ*-2.01 (m, 3H), *δ*-4.13 (m, 1H, COOCH_2_), *δ*-4.53 (m, 1H, COOCH_2_), *δ*-4.74 (dd, 1H, CHCl).

#### 2.2.2. Synthesis of mPEG-b-[PCL-g-(MEO_2_MA-co-OEGMA)]-b-mPEG and Azide End-Functionalized mPEG-b-[PCL-g-(MEO_2_MA-co-OEGMA)]-b-mPEG

The block-graft copolymer *m*PEG-*b*-[PCL-*g*-(MEO_2_MA-*co*-OEGMA)]-*b*-*m*PEG was synthesized by the method of atom transfer radical polymerization (ATRP), and the specific operations were as follows; the triblock copolymer *m*PEG-*b*-PCL-*b*-*m*PEG (412.60 mg, 0.10 mmol) was taken in a dry schlenk flask under the protection of nitrogen atmosphere. When the copolymer was completely dissolved in 5.00 mL *N*,*N*-dimethylformamide, MEO_2_MA (1.80 mL, 9.70 mmol) and OEGMA (0.13 mL, 0.30 mmol) were added to reaction system with moderated stirring. The freeze–pump–thaw operation was repeated three times in order to remove oxygen. Then, the catalyst of cuprous chloride (10.30 mg, 0.10 mmol) and 2,2′-bipyridine (31.50 mg, 0.20 mmol) was added to schlenk flask. The schlenk flask was sealed and heated in an oil bath at 70 °C. After 12 h, the system was stopped and cooled to room temperature. The viscous product was diluted with deionized water and transferred to MWCO14kDa dialysis bags for 3 days to remove impurities. After freeze-drying, the transparent colloidal *m*PEG-*b*-[PCL-*g*-(MEO_2_MA-*co*-OEGMA)]-*b*-*m*PEG was obtained.

*m*PEG-*b*-[PCL-*g*-(MEO_2_MA-*co*-OEGMA)]-*b*-*m*PEG (330.00 mg) and sodium azide (37.00 mg) were dissolved in 10.00 mL *N*,*N*-dimethylformamide in a round-bottom flask. The reaction system was stirred at room temperature for 72 h. The product was isolated by precipitated in anhydrous diethyl ether. After dilution with water, the product was transferred toMWCO14kDa dialys is bags for dialysis. Last, after freeze-drying to remove the water, a slightly white colloidal product azide end-functionalized copolymer *m*PEG-*b*-[PCL-*g*-(MEO_2_MA-*co*-OEGMA)]-*b*-*m*PEG was obtained.

#### 2.2.3. Synthesis of Cross-Linker Alkyne-Terminated P(GMA-co-MEO_2_MA-co-OEGMA)

P(GMA-*co*-MEO_2_MA-*co*-OEGMA) was synthesized by the reversible addition–fragmentation chain transfer polymerization (RAFT) method. First, the chain transfer agent CTA was synthesized according to the method in the literature [36]. CTA (36.40 mg, 0.10 mmol), MEO_2_MA (2.50 mL, 13.58 mmol), OEGMA (0.18 mL, 0.42 mmol) and GMA (0.08 mL, 0.60 mmol) were dissolved in 5.00 mL 1,4-dioxane in a dry schlenk flask under the protection of nitrogen atmosphere. The schlenk flask was placed in an ice water bath below 10 °C with moderated stirring for 40 min. Then, 20.00 mg AIBN was added to the reaction system. The schlenk flask was replaced in a 60 °C oil bath. 5 h later, the reaction system was cooled to room temperature. The product was diluted with deionized water, and then transferred to MWCO14kDa dialysis bags for dialysis. After freeze-drying to remove the water, a pale yellow product was obtained.

The above P(GMA-*co*-MEO_2_MA-*co*-OEGMA) (574.00 mg) and NaH (46.00 mg) were completely dissolved in tetrahydrofuran (THF) in a dry 10.00 mL round-bottom flask, and stirred moderately for 15 min. Then, propargyl bromide (0.17 mL) was added to the reaction system. The reaction was carried out at room temperature for 48 h, and then the product was transferred to an MWCO14kDa dialysis bags for 3 days. After freeze-drying to remove the water, the yellow product alkyne-terminated P(GMA-*co*-MEO_2_MA-*co*-OEGMA) was obtained.

#### 2.2.4. Synthesis of Cross-Linker TPOM

TPOM was synthesized according to the methods provided in the previous literature [34]. The specific procedure was KOH (15.60 g) and pentaerythritol (2.50 g) were dissolved in 30.00 mL of *N*, *N*-dimethylformamide in a dry 100.00 mL round-bottom flask. The mixture system was stirred at 5 °C for 30 min. Then, the propargyl bromide (20.80 g) was added dropwise into the reaction over 20 min. The mixture became brown, and reacted at 40 °C for 24 h. After cooling, an appropriate amount of water was added to stop the reaction, and the mixture was extracted with 50.00 mL diethyl ether. The organic layers were combined, and washed with brine and water, and then dried over MgSO_4_. Finally, the crude product was purified by the silica gel column, which using ethyl acetate-hexane (*V*/*V* = 2:8) as a mobile phase, giving an orange pure product.

#### 2.2.5. Preparation of Two Hydrogels via Click Chemistry

The azide end-function *m*PEG-*b*-[PCL-*g*-(MEO_2_MA-*co*-OEGMA)]-*b*-*m*PEG and the alkyne -terminated P(GMA-*co*-MEO_2_MA-*co*-OEGMA) in a certain molar ratio were dissolved in the deionized water, and the system was stirred. After the solution turned clear, sodium ascorbate (0.15 g) and a few drops saturated copper sulfate pentahydrate solution were added to mixture quickly. Gels appeared within 1 min. The reaction was continued until present a homogeneous solid structure gels at room temperature. The gels were rinsed with deionized water for several times to remove unreacted impurities.

The azide end-functionalized *m*PEG-*b*-[PCL-*g*-(MEO_2_MA-*co*-OEGMA)]-*b*-*m*PEG and the tetrakis (2-propynyloxymethyl)-methane (TPOM) were dissolved in *N*, *N*-dimethylformamide (DMF) in a dry 10.00 mL round-bottom flask. Then, the *N*,*N*,*N*′,*N*′,*N*″-pentamethyl diethylenetriamine (PMDETA) (17.40 mg) was added to the reaction system with moderate stirring. When the solution was clear, CuCl (7.20 mg) was added into the round-bottom flask quickly. Gels appeared within 1 min. The reaction was continued to react at room temperature for 24 h.

### 2.3. Characterization

#### 2.3.1. Nuclear Magnetic Resonance Spectroscopy (^1^H NMR)

The chemical structures of some copolymers synthesized in this work were analyzed by ^1^H NMR in CDCl_3_ using a Bruker 300 MHz spectrometer (^1^H NMR) (300 MHz AVANCE, Bruker Corporation, Karlsruhe, Germany) at room temperature. The chemical shifts are given relative to tetramethylsilane (TMS).

#### 2.3.2. Fourier-transform infrared Spectroscopy (FTIR)

The infrared spectrums of some copolymers were measured on a Bruker FTIR (Tensor 27, Bruker Corporation, Karlsruhe, Germany). Before being measured, the dried sample should be mixed with KBr, ground into a powder. and tableted.

#### 2.3.3. Gel Permeation Chromatography (GPC)

The number average molecular weight (*M*_n_), polydispersity index (*M*_W_/*M*_n_), and molecular weight distribution (PDI) of the copolymers were recorded by gel permeation chromatography (GPC) (Breeze, Water, Milford, MA, USA). In method, tetrahydrofuran is both mobile phase and solvent, and a series of different molecular weight PEG aqueous (2 mg·mL^−1^) were prepared as standard samples. Then test the copolymer’s GPC trace and determine the *M*_n_, *M*_W_, and PDI. All of the sample aqueous in above were filtered with a 0.25-μm oil phase filter before being measured.

### 2.4. Water Solubility and Temperature Sensitivitys

The water solubility and temperature sensitivity were predetermined by taking digital photos of the copolymer aqueous (2.00 mg/mL) in a transparent glass vial at 25, 35 and 45 °C respectively.

In addition, the water solubility and temperature sensitivity of the block-graft copolymers were illustrated by the transmittance measured with a UV–Vis spectroscopy (TU-1901, Beijing Purkinje General Instrument Corporation, Beijing, China).The transmittance of a series copolymers aqueous (2.00 mg/mL) was measured by UV–Vis spectrophotometer at different temperature, and then the correlation curve was drawn according to the transmittance change with temperature; the inflection points of the curve is the low critical solution temperature (LCST) of the different component block-graft copolymers. We configured the solutions at 0.50, 1.00, 2.00, 4.00, and 8.00 mg/mL tBG4 aqueous solutions, respectively, and then the transmittance change of the different solutions from 25 to 50 °C was measured to obtain the low critical solution temperature (LCST).

### 2.5. Micelle Properties of Polymers

Determination of critical micelle concentration (CMC): (1) Surface tension is one of the methods to determine the critical micelle concentration value of the copolymers. This experiment measured the surface tension of the copolymer solutions with the surface tension meter (DCAT21, Germany Dataphysics Corporation, Stuttgart, Germany). (2) Fluorescent probe technology is another way to determination the critical micelle concentration. The measurement was performed with a fluorescence spectrophotometer (PELS55, America Perkin Elmer Corporation, Waltham, MA, USA), and used pyrene as a steady-state hydrophobic fluorescent probe. The peak intensities ratios of the third peak (*I*_3_, 384 nm) to the first peak (*I*_1_, 373 nm) of the emission spectra were recorded, and then plotted as*I*_3_/*I*_1_ ratio and log*c* curves; the critical micelle concentration (CMC) value is defined as the turning point of the linear regression lines on plots.

Dynamic light scattering (DLS): The particle size distribution of the block-graft copolymers aqueous was gauged by a dynamic light scattering (DLS) instrument ZS 90 (Otsukaelectronics, Osaka, Japan), which has a 4 mW He–Nesolid-state laser. Each sample solution (1.00 mg·mL^−1^) was filtered with a 0.45-μm water phase filter before measurement. The laser wavelength was 633 nm, the scattering angle was 90°, and the temperature range was from 25 to 50 °C.

Transmission electron microscopy (TEM): The micelle morphology was observed by JEOL JEM-2100 transmission electron microscopy (Tokey, Japan) with accelerating voltage 200 kV. The samples was prepared by placing a drop of 1.00 mg·mL^−1^ copolymers solution on a carbon-coated copper grid (200 mesh), stained with 1.00 wt % phosphotungstic acid solution, and then dried in the vacuum at 25 and 35 °C, respectively.

### 2.6. Sol–gel Transition

The gel-to-sol transition of the copolymer solutions was measured by the test vial inverting method. The different mass copolymers were dissolved in water in a vial and stirred for 2 days at room temperature. Then, we placed vials of copolymer solutions in a water bath from 20 °C to 50 °C and waited until the solutions were equilibrated at every temperature for 20 min. The vials were inverted for 5 s to observe the states of these solutions, and the temperature at which the copolymer solutions changed from a flow state to a stationary state was recorded.

### 2.7. SEM Analysis of Gel

The surface morphology of the two hydrogels was taken by an environmental scanning electron microscope (Quanta 200). The gels were immersed in distilled water to swell equilibrium, and then were placed in liquid nitrogen quickly for preserved the gels morphology. When dried by a freeze dryer, the gels’ microscopic morphology was measured at a high vacuum mode. In order to increase the conductivity of the sample, the gels were sprayed with gold.

## 3. Results and Discussion

### 3.1. Synthesis of the Copolymers and Hydrogels

In this study, the triblock-graft copolymers *m*PEG-*b*-[PCL-*g*-(MEO_2_MA-*co*-OEGMA)]-*b*-*m*PEG were synthesized by the combination of ROP and ATRP. First, the *m*PEG-*b*-PCL diblock copolymers were synthesized via ROP of *ε*-Caprolactone and at 120 °C for 12 h, the sn(Oct)_2_ was the catalyst. Moreover, the *m*PEG-*b*-PCL-*b*-*m*PEG triblock copolymers were produced by couple the *m*PEG-*b*-PCL diblock copolymer with HMDI at 60 °C for 6 h. Then, the *m*PEG-*b*-[PCL-*g*-(MEO_2_MA-*co*-OEGMA)]-*b*-*m*PEG triblock-graft copolymers were synthesized by ATRP and azide triblock-graft copolymer to obtain azide en-functionalized triblock-graft copolymer. Under the premise of the triblock-graft copolymer generated, the cross-linker alkynyl P(GMA-*co*-MEO_2_MA-*co*-OEGMA) was produced by RAFT, and another cross-linker agent TPOM was synthesized by organic reaction. Finally, the azide end-functionalized triblock-graft copolymers were reacted with the two cross-linkers via click chemistry separately, obtained two hydrogels under different conditions. As shown in Scheme 1.

Appendix A represent the infrared spectra spectrum and nuclear magnetic spectrum of the copolymers; the absorption peak or vibration peak of the special functional group has been marked.

Appendix A shows the infrared spectrum of the azide en-functionalized triblock-graft copolymer, the alkyne-terminated P(GMA-*co*-MEO_2_MA-*co*-OEGMA), the TPOM, and the two gels. Appendix A, as with Appendix A, is the infrared spectrum of the azide en-functionalized triblock-graft copolymer. Appendix A represents the alkyne-terminated P(GMA-*co*-MEO_2_MA-*co*-OEGMA). Appendix A is the infrared spectrum of TPOM, the 2100 and 3280 cm^−1^ peaks were the shrinkage vibration peaks of C≡C and C–H on C≡CH. Appendix A represents the infrared spectrum of the two gels, respectively, the infrared spectrum of the gels no longer shows the characteristic functional groups of the reactants of the cross-linkers, indicating that the azide en-functionalized triblock-graft copolymer reacts well with both cross-linkers.

### 3.2. GPC Characterization of the Copolymers

In the step of *m*PEG-*b*-PCL-*b*-*m*PEG triblock copolymer synthesis, the total amount of *ε*-Caprolactone and *α*-Chloro-*ε*-Caprolactone added to the reaction remained unchanged, while the ratios of the *ε*-Caprolactone and the α-Chloro-*ε*-Caprolactone were 8:2, 7:3, 6:4, and 5:5, respectively. The difference in the content of the characteristic functional group Cl determines the difference in the total amount of the grafted MEO_2_MA and OEGMA copolymer on the triblock-graft copolymer. Therefore, the molecular weights of the copolymers also were differs. Figure 1 shows the GPC trace of the four polymers. All of the curves are single peaks, and with narrow distribution, indicating that the method used in the experiment is indeed advantageous for the synthesis of copolymers with narrow molecular weight distribution. In addition, with the increase of the theoretical molecular weight of the copolymers, the GPC trace grew gradually broader. This illustrated that the larger the copolymer’s molecular weight designed, the more the product distribution dispersed. Detailed information of each copolymer is shown in Table 1.

### 3.3. Relationship between Conversion Rate and Molecular Weight

As can be seen from the Table 1 and Figure 2a, the conversion rate of the copolymers increased progressively with reaction time. When the reaction time is less than 12 h, the conversion rate increases rapidly, while the reaction conversion rate increases slowly after 12 h. In addition, the effect of conversion on Mn and PDI has been shown in Figure 2b, and, as can be seen from the figure, the molecular weight of the copolymer increases continuously with the increases of reaction conversion, and the PDI of the copolymer is also increases at the same time. 

### 3.4. Water Solubility and Temperature Sensitivity

The hydrophilic *m*PEG segment of the triblock copolymers *m*PEG-*b*-PCL-*b*-*m*PEG was limited, and the hydrophobic PCL chain determined that the water solubility of the copolymer was relatively poor [30]. Once the triblock copolymer *m*PEG-*b*-PCL-*b*-*m*PEG was added to deionized water, the solution became cloudy. While the P(MEO_2_MA-*co*-OEGMA) graft chains have improved the water solubility of the *m*PEG-*b*-[PCL-*g*-(MEO_2_MA-*co*-OEGMA)]-*b*-*m*PEG triblock-graft copolymers. This is because the hydrophilic oligo(ethylene glycol) graft chain on the copolymers formed hydrogen bonds with water molecules at room temperature. In addition, the PEG analog copolymers formed by the MEO_2_MA and OEGMA have unique temperature sensitivity. Therefore, these two segments make the copolymer have temperature sensitivity [31]. Figure 3 is the photographs of the triblock-graft copolymers (tBG) solutions at 25, 35 and 45 °C. The tBGs solutions were clear at 25 °C, while gradually cloudy with temperature rise.

The LCST of copolymers were with a UV–Vis spectroscopy. As shown in Figure 4a, the transmittance from tBG1 to tBG4 was gradually increased at room temperature, which was because the amount of the characteristic functional group Cl was increased, in other words, the density of the hydrophilic copolymer P(MEO_2_MA-*co*-OEGMA) segment gradually increases from the tBG1 to tBG4. In addition, as can be seen from the figure, the transmittance of the four copolymer aqueous solutions began to decrease sharply after 35 °C, indicating that the LCST of the tBGs solutions was ~35 °C. Beyond 40 °C, the transmittance of tBG1 became steady, but the others solutions’ transmittance was 0%, this illustrated that the content of MEO_2_MA and OEGMA was one of the important factor that affect the temperature sensitivity of the copolymer solutions at the same concentration.

Figure 4b is a graph showing the effect of temperature on tBG4 solutions transmittance with different concentration, and Figure 4c is the LCST curve of the tBG4 solutions with different concentration. Combining the two graphs, it can be seen that the LCST of the tBG4 copolymer solutions gradually decreases as the concentration increases from 0.50 to 8.00 mg/mL. This indicated that solution concentration was also an important influence factor when studying copolymer solutions; a similar conclusion has been reported before [37].

### 3.5. Critical Micellization Concentration Determination (CMC)

The copolymer molecules are present in a single molecule state when the amphiphilic copolymer solution concentration is very low. Until the concentration increases to a certain value, the copolymer molecules self-assemble into nanoscale core–shell micelles, so this certain concentration value is the critical micelle concentration of the copolymer solution. There are many methods to determine critical micelle concentration. In this study, two experimental methods were used to search the critical micelle concentration: a fluorescence spectrometer and a surface tension meter.

Figure 5a shows that the CMC of the tB4 solution was 8.30 × 10^−3^ mg/mL. Figure 5b shows that the CMC of tBG2, tBG3, and tBG4 solutions were 2.37 × 10^−2^, 5.99 × 10^−2^, and 8.41 × 10^−2^ mg/mL, respectively, and these values were Three to ten times greater than the CMC value of the tB4 solution. This explains the CMC value of the block-graft copolymer increases as the P(MEO_2_MA-*co*-OEGMA) chain. Because the CMC value of the amphiphilic block-graft copolymer aqueous depends on the length of the hydrophilicity and hydrophobic block, that is, the more of the hydrophilic chain P(MEO_2_MA-*co*-OEGMA) added, the larger the CMC value of the block-graft copolymer when aqueous [38,39].

Figure 5c is a graph showing the surface tension at different concentrations of tBG2, tBG3, and tBG4 copolymers solutions measured by the surface tension method, and the turning point is the CMC values of the copolymers. Before the turning point, the surface tension value decreases rapidly with the increase of the solution concentration. At the turning point, the block-graft copolymer self-assembly forms core–shell micelles in the solution. After the turning point, the surface tension hardly changes with the increase of the solution concentration. The results measured by this method are close to the CMC value measured by the fluorescent probe, and the CMC value is gradually increasing from the tBG2–tBG4 solution under the same conditions. It also demonstrates that under certain conditions, the better hydrophilicity of the copolymer is the larger CMC is.

### 3.6. Particle Size of Copolymer Micelles

Figure 6a is a graph showing the copolymers’ particle size distribution measured by the dynamic light scattering instrument at room temperature. It can be seen from the figure that the particle size of the tBG1–tBG4 copolymers are centralized at 60–70 nm under the same conditions. At the same time, the micelle particle size distribution range expands slightly from tBG1 to tBG4, which is because the density of the graft chain P(MEO_2_MA-*co*-OEGMA) was increased. Theoretically, the more the P(MEO_2_MA-*co*-OEGMA) chains in the block-graft copolymer, the stronger the interaction of the hydrophilic groups at the shell portion of the micelles, so the micelle size should smaller too. The result of the graph is probably due to unevenness in the particle size of the micelles that is produced during the synthesis [40].

Figure 6b shows the particle size distribution of the tBG4 copolymer micelles at 25 and 37 °C. It can be seen that the particle size of the micelles is mainly distributed around 60 nm at 25 °C, while increases to about 200 nm when temperature arrival to 37 °C. This was because the tBG4 block-graft copolymer self-assembles into a core–shell micelle in aqueous solution at 25 °C, and the temperature responsive graft chain P(MEO_2_MA-*co*-OEGMA) segments formed hydrogen bonds with water molecules, all of the hydrophilic chains existed with an extended state, so the micelles are present in the solution with a stable structure. But, when the temperature is increased to 37 °C, the hydrogen bonds between the graft chain P(MEO_2_MA-*co*-OEGMA) and the water molecules become weaker [41], the P(MEO_2_MA-*co*-OEGMA) chain collapses outside the micelles in a short time frame, and a large amount of P(MEO_2_MA-*co*-OEGMA) graft chain adheres to the micelles making the micelles are unstable in solution, which made the micelle aggregates coexist with the larger particle size single micelles. This can be reflected in the increased turbidity of the copolymer solution [33].

Transmission electron microscopy is one of the most common methods for studying the morphology of copolymer micelles. This method can not only clearly observe the morphology of the micelles, but also measure the particle size of the micelles. In order to intuitively show the micelles of the copolymer in aqueous solution, taken tBG4 as an example, measured the transmission electron micrographs of the tBG4 copolymer micelles at 25 and 37 °C, as shown in Figure 6c. The morphology of the micelles can be clearly seen from the figure. According to the water solubility research before, the copolymer aqueous solution was very homogeneous at 25 °C; therefore, the prepared micelles were evenly dispersed on the copper mesh at this temperature. In this situation, the particle diameter of the micelles was about 50 nm. The copolymer aqueous solution began to become cloudy when preheated to 37 °C; this is because the hydrogen bond between the graft chain P(MEO_2_MA-*co*-OEGMA) with the water molecules began to be destroyed, the micelles sample prepared in this case can be observed the larger particle size, and the particle diameter were 160 nm, but these micelles no longer had a regular spherical shape, this was probably because some of the micelles were stacked together. Comparing Figure 6a with Figure 6b, it can be found that the particle size of the micelles measured by the dynamic light scattering instrument is slightly larger than the micelles diameter observed by the transmission electron microscope, this is because the value of the sample particle size observed by TEM was in a dry state, and the blocks were contracted, while the value of the sample particle size observed by DLS was measured the block-graft copolymers’ solutions, in which the copolymer chains were in a stretched state.

### 3.7. Sol–Gel Transition

Figure 7a recorded the sol–gel transition of the different concentrations tBG4 copolymer aqueous solutions. Then, we placed all vials of the copolymer solutions in the water bath, until the solution was equilibrated at every temperature. At each temperature, the sample was equilibrated for 15 min, then the vials were inverted for 5 s to examine whether the copolymer was a flowing liquid or an immobile micelle gel under its own weight [30]. As can be seen from the figure, the sample solutions were still in a flowing state before 25 °C after the temperature was risen to 30 °C, and copolymer solutions with 25% (wt) and 20% (wt) concentration formed a gel and were inverted for 15 min; the gel flow light. Subsequently, the copolymer solutions with 10% (wt) and 15% (wt) concentration were also converted into gel at 30 °C, while the two samples were dehydrated directly, and appeared white flocculent substance at 35 °C. As the temperature continues to rise, each component copolymer gel began to dehydrate. This process illustrated that the sol–gel transition temperature of the copolymer aqueous solution was decreases with the increase of the concentration.

Figure 7b is the sol–gel transition phase diagram, which was plotted according to the copolymer solution state change at different temperature in order to directly show the relationship between the concentration of the solutions and the sol–gel transition temperature. When the concentration reached a certain level, the amount of micelles in the copolymer aqueous solution was very large, the branches out the core–shell structure of micelles were in contact with each other, and form a plurality of cross-link points which can cross-link to constitute the gel. The greater the concentration of the copolymers solution, the more cross-link points provided, and at the same time, the induced temperature was lower. Continue to increase the temperature after the copolymer solutions transformed into a gel, the graft chains of copolymer micelle collapsed, the dehydration in the core was intensified, the cross-linking point were destroyed. We can see that the collapsed dehydrated micelles were partially viscous, the water was separated out of the micelle, and the copolymer aqueous solution is converted from the gel state to white precipitation.

The sol–gel transition of the copolymer solution based on the cross-linking point, which was formed by the weak interaction between the outer branches of the micelle core; it is a physical cross-linking, so the sol–gel transition process is reversible, the copolymer aqueous solution can achieve a reversible transformation of the sol–gel depending on the temperature rise and fall [42], as shown in Figure 7c.

### 3.8. SEM Analysis of Gel

The morphology of the two dry gels was observed under a scanning electron microscope, as shown in the Figure 8. Both gels have the relatively obvious network structure. Figure 8a,b show the scanned photographs of the gel formed by the azide end-function *m*PEG-*b*-[PCL-*g*-(MEO_2_MA-*co*-OEGMA)]-*b*-*m*PEG copolymer and the alkyne-terminated P(MEO_2_MA-*co*-OEGMA) at 25 and 40 °C, respectively. Figure 8c,d show the scanned photographs of the gel formed by the azide end-functionalized copolymer *m*PEG-*b*-[PCL-*g*-(MEO_2_MA-*co*-OEGMA)]-*b*-*m*PEG with the TPOM at 25 and 40 °C, respectively. Comparing the gel pore sizes in Figure 8a–d, it was found that the pore size of the first gel was slightly smaller than the second one. This is because the observed porous structure of freeze-dried hydrogels were due to the phase separation of the gels during rapid freezing and subsequent removal of the water by sublimation which left voids in place where the water previously occupied, and the increased cross-link density results in the second gel faster phase separation during freezing [43]. Then, comparing the changes of the two gels between 25 °C with 40 °C, it can be seen that the gel was in a shrinking state at 40 °C, and the pore size of three-dimensional network become smaller. The reason for this is that the hydrophobic interactions of PCL–PEG chains at a higher temperature (*T* > LCST). The SEM images of the hydrogels at different temperature certified that two gels are temperature-sensitive [44].

## 4. Conclusions

The novel hydrogel is composed of the *m*PEG-*b*-[PCL-*g*-(MEO_2_MA-*co*-OEGMA)]-*b*-*m*PEG block-graft copolymer with the alkynyl P(GMA-*co*-MEO_2_MA-*co*-OEGMA) and TPOM via click chemistry. The water-soluble and temperature-sensitive *m*PEG-*b*-[PCL-*g*-(MEO_2_MA-*co*-OEGMA)]-*b*-*m*PEG block-graft copolymer was synthesized by ROP and ATRP. The alkynyl P(GMA-*co*-MEO_2_MA-*co*-OEGMA) was synthesized by RAFT and has good water solubility and temperature sensitivity. The LCST of the block-graft copolymer is designed within the normal physiological temperature of the human body. The results of dynamic light scattering and transmission electron microscopy specifically show that the copolymer exist with nanometer size micelles in water, and the state of the micelles can affected by temperature. Sol–gel transition illustrated that the copolymer aqueous solution can achieve a reversible transformation of the sol–gel depending on the temperature rise and fall. After form the gel via click chemistry, the internal three-dimensional network morphology of the gel measured by scanning electron microscopy at different temperatures certified that the hydrogel owned temperature sensitivity. The gels are expected to be used as a material for tissue damage repair. Injecting it with a mixture of growth factors into the human body in-situ click to form a chemical gel can alleviate the pain of surgery.

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
