# Peer review of "Synthesis of Thermo-Responsive Block-Graft Copolymer Based on PCL and PEG Analogs, and Preparation of Hydrogel via Click Chemistry"

_polymers, 2019, doi:10.3390/polym11050765_

Round 1
Reviewer 1 Report
The authors present the preparation of hydrogel via click reaction using the thermo-responsive block-graft copolymer based on PCL and PEG. Although the synthetic process is relatively complicated, it still could contribute to this field. Thus, I would suggest a major revision with following comments:
(1) The authors leave a simple table of polymer samples used in this study. I think this table should be moved to the manuscript with more information, such as conversion, reaction time, …etc.
(2) The authors should provide more details of synthetic process and characterization. For example, the conversion versus reaction time, the plots of Mn and PDI versus conversion. Regarding to the GPC measurement, no GPC trace has been shown. In addition, the GPC was calibrated by polystyrene standard but the samples are PCL and PEG based copolymers. The GPC results need to be recalculated.
(3) Generally, the fonts in figures are too small.
(4) Section 3.1 describes the results of synthesis and characterization but all the data are in supporting information, if those data are so important, they should be moved to manuscript. If they are not important, this paragraph should not be so long.
(5) There are 4 triblock-graft copolymers but in the property tests, sometime the authors show 2 tBG, sometime show 1 tBG, sometime use tBG3 as example, sometime use tBG4 as example. What’s the reason for sample selection? The authors should’ve mentioned how they select the tBG sample and consistently use the same samples for all property test.
(6) In figure 6, the authors used the wall thickness and hole size to demonstrate the thermo-responsiveness. However, the change of temperature supposes to be able to affect these two factors for the preparation of porous materials, no matter it’s thermo-responsive materials or not. The authors should’ve provide the evidence and ref. to confirm that the wall thickness and hole size can be used to demonstrate the thermo-responsiveness.
(7) the effective digit should be reasonable and consistent.
(8) The English writing should be checked.
Author Response
Response to Reviewer 1 Comments
Thank you very much for your comments concerning our manuscript. Our responses to several comments are listed below:
Point 1: The authors leave a simple table of polymer samples used in this study. I think this table should be moved to the manuscript with more information, such as conversion, reaction time, …etc.
Response 1: As the reviewer suggested that, we moved the table of polymer samples to the manuscript and added some information about reaction time and conversion in it.
Point 2: The authors should provide more details of synthetic process and characterization. For example, the conversion versus reaction time, the plots of Mn and PDI versus conversion. Regarding to the GPC measurement, no GPC trace has been shown. In addition, the GPC was calibrated by polystyrene standard but the samples are PCL and PEG based copolymers. The GPC results need to be recalculated.
Response 2: We are very sorry for our negligence in measuring GPC. We used a series of PEGs with different molecular weights as standard samples, re-measured the GPC of the samples. We added details information as the reviewer suggested, analyzed the relationship between conversion rate and reaction time, and the connection of Mn and PDI versus conversion.
Point 3: Generally, the fonts in figures are too small.
Response 3: We are very sorry for the wrong format. We have made an adjustment.
Point 4: Section 3.1 describes the results of synthesis and characterization but all the data are in supporting information, if those data are so important, they should be moved to manuscript. If they are not important, this paragraph should not be so long.
Response 4: We have made changes according to your comments.
Point 5: There are 4 triblock-graft copolymers but in the property tests, sometime the authors show 2 tBG, sometime show 1 tBG, sometime use tBG3 as example, sometime use tBG4 as example. What’s the reason for sample selection? The authors should’ve mentioned how they select the tBG sample and consistently use the same samples for all property test.
Response 5: We are very sorry for our negligence. We have made corrections, the single test sample is tBG4, this is because tBG4 shown more obvious temperature responsive and water solubility according to the theory designed.
Point 6: In figure 6, the authors used the wall thickness and hole size to demonstrate the thermo-responsiveness. However, the change of temperature supposes to be able to affect these two factors for the preparation of porous materials, no matter it’s thermo-responsive materials or not. The authors should’ve provide the evidence and ref. to confirm that the wall thickness and hole size can be used to demonstrate the thermo-responsiveness.
Response 6: We are very sorry for our inappropriate expression. In the SEM diagram of the gel, the thickness of the three-dimensional network is related to the adequacy of the reaction, and the size of the aperture is related to the crosslinking density, and the contracted three-dimensional network of the gel at high temperature is due to the hydrophobic effect of PCL-PEG block. So, the comparison of SEM photos of gels at different temperatures shows that gels have temperature sensitivity. This part has been attached to the corresponding reference.
Point 7: the effective digit should be reasonable and consistent.
Response 7: We are very sorry for the wrong writing format. We have corrected.
Point 8: The English writing should be checked.
Response 8: We are very sorry for our incorrect writing. We made further changes according to your comments.
Reviewer 2 Report
The manuscript by Liu et al described the synthesis and characterization of novel polymer PCL-PEG-(graft POEGMA). Authors provided adequate synthesis and characterization for synthesized polymers and introduced the thermo-responsiveness of the polymer solution.
Overall the manuscript is well written and organized. I would recommend it published as it is in Polymers. It is recommended but not necessary that authors introduce some potential applications in the discussion part. But this might be some interesting work in future.
Author Response
Response to Reviewer 2 Comments
I have supplemented the potential application of the gel in the discussion section according to your comments. Thank you very much for your comments concerning our manuscript.
Round 2
Reviewer 1 Report
I am satisfied with the revised version.